# Learning to Match Unpaired Data with Minimum Entropy Coupling

**Mustapha Bounoua** [1][2]   **Giulio Franzese** [2]   **Pietro Michiardi** [2]

## Abstract

Multimodal data is a precious asset enabling a variety of downstream tasks in machine learning. However, real-world data collected across different modalities is often not paired, which is a significant challenge to learn a joint distribution. A prominent approach to address the modality coupling problem is Minimum Entropy Coupling (MEC), which seeks to minimize the joint Entropy, while satisfying constraints on the marginals. Existing approaches to the MEC problem focus on finite, discrete distributions, limiting their application for cases involving continuous data. In this work, we propose a novel method to solve the continuous MEC problem, using well-known generative diffusion models that learn to approximate and minimize the joint Entropy through a cooperative scheme, while satisfying a relaxed version of the marginal constraints. We empirically demonstrate that our method, DDMEC , is general and can be easily used to address challenging tasks, including unsupervised single-cell multi-omics data alignment and unpaired image translation, outperforming specialized methods.

## 1. Introduction

Nowadays, multimodal data is pervasive thanks to advances in data collection technologies and the crucial need for systems that can learn from the diversity of real-world phenomena. Healthcare, for example, is a domain where patient data often spans electronic health records, radiological images, genetic data, and wearable sensor outputs (Kline et al., 2022; Acosta et al., 2022). Autonomous systems rely on a suite of sensors, including LiDAR, cameras, and ultrasonic sensors, to navigate environments effectively (Caesar et al., 2020; Gu et al., 2023; Franchi et al., 2024). Scientific

disciplines, such as astronomy and geoscience, employ multimodal datasets combining spatial, spectral, and temporal data to understand complex systems (Srivastava et al., 2019; Zhang et al., 2024a; Šošić Klindžić et al., 2024).

Modeling multimodal data allows for a more comprehensive understanding, reflecting the inherently multi-faceted nature of the real world. Recent works in representation learning (Radford et al., 2021; Lu, 2023; Manzoor et al., 2023; Chen et al., 2023), the study of multivariate systems (McSharry et al., 2024; Liang et al., 2023; Bounoua et al., 2024b), generative modeling (Rombach et al., 2022; Tang et al., 2023b;a; Bounoua et al., 2024a; Esser et al., 2024), and multimodal conversational agents (Li et al., 2023; Liu et al., 2023; Shukor et al., 2023; Xue et al., 2024; Wu et al., 2024), are few examples to illustrate the fervent effort in the machine learning community to address and exploit multimodality. However, the intrinsic complexity of multimodal data introduces several challenges that hinder their application in machine learning research. Modality heterogeneity complicates and sometimes impedes geometric comparisons, requiring for example learning a mapping from ambient to latent spaces (Rombach et al., 2022; Tang et al., 2023b; Liu et al., 2023; Bounoua et al., 2024a) or stringent assumptions (Liang et al., 2022; Xia et al., 2023; Dong et al., 2024; Ibrahimi et al., 2024; Zhang et al., 2024b). Alignment across modalities at spatial, temporal or semantic levels is another challenge, which calls for costly pre-processing steps such as synchronization (Hanchate et al., 2024; Chen et al., 2024; Scirè, 2024; Martin-Turrero et al., 2024).

The major roadblock we address is that of paired multimodal data, which is an underlying assumption in many works in the literature (Radford et al., 2021; Rombach et al., 2022; Liu et al., 2023; Li et al., 2023; Bounoua et al., 2024a). Paired data – for a given sample, all its various modalities are available – is either expensive, difficult to obtain, or sometimes impossible. For example, in genetic research, data is inherently unpaired due to the nature of the data acquisition process, such as single-cell RNA sequencing data, where measurements destroy the original cells (Kester & van Oudenaarden, 2018; Chen et al., 2019; Schiebinger et al., 2019). Similarly, matching image data from different domains is a challenging endeavor when paired data is missing, which calls for specialized methods (Zhu et al., 2017; Huang et al., 2018; Pang et al., 2021; Sasaki et al., 2021;

---

[1]Ampere Software Technology, France [2]Department of Data Science, EURECOM, France. Correspondence to: <mustapha.bounoua@eurecom.fr>.

*Proceedings of the 42nd International Conference on Machine Learning*, Vancouver, Canada. PMLR 267, 2025.

Yang et al., 2023; Sun et al., 2023; Xie et al., 2023).

In this work, we study the problem of unpaired multimodal data through the lens of *coupling*, a fundamental problem in probability theory, that aims at *determining the optimal joint distribution of random variables given their marginal distributions*, with early attempts at solving it dating back to the work by Fréchet (1951). The pairing problem belongs to a broad class of methods (Den Hollander, 2012; Lin et al., 2014; Benes & Stepán, 2012; Yu & Tan, 2018): some cast it through the lens of information-theoretic quantities, where optimality is defined in terms of Entropy minimization or Mutual Information maximization, others focus on Optimal Transport (OT) (Villani, 2009; Peyré & Cuturi, 2019), where optimality is defined as minimizing the expected value of a transport cost over the joint distribution. Our focus is the Minimum Entropy Coupling (MEC) problem, which aims at finding the joint distribution with the smallest Entropy, given the marginal distribution of some random variables. Recent applications include entropic causal inference (Kocaoglu et al., 2017; Javidian et al., 2021; Compton, 2022), communication systems (Sokota et al., 2022), steganography (de Witt et al., 2022), random number generation (Li, 2021), dimensionality reduction (Cicalese et al., 2016; Vidyasagar, 2012), lossy compression (Ebrahimi et al., 2024), and multimodal learning (Liang et al., 2024).

While the MEC problem is known to be NP-Hard (Vidyasagar, 2012; Kovačević et al., 2012), the literature contains many approximation and greedy algorithms (Painsky et al., 2013; Kovacevic et al., 2013; Cicalese et al., 2016; Li, 2021), and theoretical studies about the approximation qualities of such approaches (Cicalese et al., 2017; 2019). Nevertheless, the vast majority of prior work on the MEC problem focus on discrete distributions: instead, we consider the continuous variant of MEC, and propose a flexible and general solution to the coupling problem for arbitrary, continuous distributions. The MEC problem for continuous random variables is much more complex than its discrete counterpart, and can be ill-defined in certain cases due to the properties of differential Entropy and the challenges inherent to continuous distributions living in an infinite dimensional space.

The gist of our method is to consider a parametric class of joint distributions, which we reinterpret as conditional generative models, with additional terms to steer adherence to marginal constraints. Then, the MEC problem requires access to the conditional Entropy, which we rewrite as log-likelihood. Crucially, our method exploits two specular generative models, which cooperate to minimize the joint Entropy, while approximately satisfying the marginal constraints. Our approach materializes as two denoising diffusion probabilistic models (Ho et al., 2020), which we first pre-train on marginal distributions, and then fine-tune

according to reward functions, following an alternating optimization process. In summary, our contributions are:

- We propose an approximation of the MEC problem for arbitrary, continuous distributions, which is general, and that does not require stringent assumptions on the marginal distributions, nor the definition of geometric cost functions (Section 2).

- We present a practical implementation of our method (Section 3), that relies on generative models, that interact through a cooperative scheme aiming at optimizing an information-theoretic cost function related to the Entropy of the joint distribution. Our training procedure overcomes numerical instabilities and degenerate solutions by relying on the application of soft marginal constraints, as well as the natural approximation stemming from a finite-capacity denoising model.

- We illustrate the benefits and performance of our method on two important use cases (Section 4). First, we solve the coupling problem between incomparable spaces with a single-cell multi-omics dataset, where we compare our method to state-of-the-art alternatives that rely on OT. Second, we focus on unsupervised image translation between uncoupled pairs, and compare against state of the art.

## 2. Problem Formulation

Given two random variables $X \in \mathcal{X}$ and $Y \in \mathcal{Y}$ with marginal probability distributions $p_X(x)$ and $p_Y(y)$ respectively, we consider a *parametric* space $\mathcal{P}^\theta = \{p_{XY}^\theta(x, y)\}$ of *joint* distributions over the space $\mathcal{X} \times \mathcal{Y}$, with induced marginal distributions $p_X^\theta(x), p_Y^\theta(y)$ (where $p_X^\theta(x) \triangleq \int_{\mathcal{Y}} p_{XY}^\theta(x, y) \, dy$ and similarly for $p_Y^\theta(y)$). The MEC problem between the two original distributions $p_X(x)$ and $p_Y(y)$ consists in finding a joint distribution $p_{XY}^\theta(x, y)$ such that i) the induced marginal distributions $p_X^\theta(x), p_Y^\theta(y)$ match them either exactly or approximately and ii) the joint distribution is the one with minimal entropy (Kovacevic et al., 2013; Cicalese et al., 2017; 2019). The constraints over the search space $\mathcal{P}^\theta$ are referred to as *marginal contraints*

**Definition 2.1.** A joint distribution $p_{XY}^\theta(x, y)$ from $\mathcal{P}^\theta$ is said to be an *exact* coupling iff

$$p_X^\theta(x) = p_X(x), p_Y^\theta(y) = p_Y(y). \tag{1}$$

In general, exact coupling is not possible (nor wanted, to avoid overfitting) and the goodness of the solution in terms of marginal constraints is approximated through some distance function between the induced and original distributions, e.g. using the Kullback-Leibler divergence $\mathbb{KL}\left[p_X^\theta \parallel p_X\right] \triangleq \mathbb{E}_{x \sim p_X^\theta}\left[\log \frac{p_X^\theta}{p_X}(x)\right]$. Then, we define

the MEC problem with *soft* constraints as follows

$$\min_{\theta} \mathbb{H}(p_{XY}^{\theta}) + \lambda_X \mathbb{KL}\left[p_X^{\theta} \parallel p_X\right] + \lambda_Y \mathbb{KL}\left[p_Y^{\theta} \parallel p_Y\right], \tag{2}$$

where the entropic term is defined as $\mathbb{H}(p_{XY}^{\theta}) \triangleq -\mathbb{E}_{x,y \sim p_{X,Y}^{\theta}}\left[\log p_{XY}^{\theta}(x,y)\right]$.

Previous work have mainly focused on the exact MEC in discrete settings, where $p_X$ and $p_Y$ have a finite or countably infinite number of outcomes. Exact solution in such settings is known to be NP-Hard (Vidyasagar, 2011; Kovacevic et al., 2013). Under our assumption of continuous spaces the problem is more complex. Exact matching is not generally possible due to the finite complexity of the parametric family $\mathcal{P}^{\theta}$, since in general the distributions $p_X, p_Y$ live in infinite dimensional spaces. Rather than a limitation, enforcing limited complexity is helpful to avoid degenerate, deterministic joint probabilities (e.g. $p_{XY}^{\theta}(x,y) = \delta(y - g(x))p_X(x)$, where $g(\cdot)$ is any mapping which guarantees exact coupling), which would induce infinite joint entropy.

Interestingly, the MEC problem has an intuitive interpretation connected to the problem of Mutual Information maximization. Indeed, $\mathbb{I}(p_{XY}^{\theta}) \triangleq -\mathbb{H}(p_{XY}^{\theta}) + \mathbb{H}(p_X^{\theta}) + \mathbb{H}(p_Y^{\theta})$ and in the exact matching scenario $\mathbb{H}(p_X^{\theta}) = \mathbb{H}(p_X), \mathbb{H}(p_Y^{\theta}) = \mathbb{H}(p_Y)$. In other words, whenever the marginal constraints are satisfied with reasonable quality, the MEC problem is a good approximation of the information maximization problem.

Early instances of the coupling problem express it through the lenses of OT (Monge, 1781; Kantorovich, 1942). In the simplest (albeit rich and interesting) scenario, the goal is to minimize the transportation cost between distributions $\mathbb{E}_{x,y \sim p_{X,Y}^{\theta}}[||x - y||^2]$, with the implicit assumption of $\mathcal{X} = \mathcal{Y} = \mathbb{R}^N$ and under the requirement of exact matching (corresponding to $\lambda_X = \lambda_Y = \infty$). Several interesting extensions, including additional constraints on the joint distribution such as geometry or structural constraints, lead to tailor-made approaches (Villani, 2009; Peyré & Cuturi, 2019). Other than the trivial relaxation of constraints from exact to approximate, a particularly useful extension concerns the *entropy-regularized* version of this problem, where the cost function is complemented by the entropic term $\mathbb{H}(p_{XY}^{\theta})$. Although MEC is fundamentally different than OT, a link between the two clearly exists. However, a straightforward comparison is not possible, as the entropic term enters the respective minimization problems with different signs. Minimizing $\mathbb{H}(p_{XY}^{\theta})$ directly over other geometric costs (like the euclidean norm considered in OT) has several advantages in terms of generality, as it does not require geometrically comparable spaces $\mathcal{X}$ and $\mathcal{Y}$.

## 3. Methodology

Consider two random variables in continuous domains, $X \in \mathcal{X}$ and $Y \in \mathcal{Y}$. We begin by considering a parametric class for the joint distribution expressed as $p_{X,Y}^{\theta}(x,y) = p_{X|Y}^{\theta}(x|y)p_Y(y)$, such that the joint entropy $\mathbb{H}(p_{X,Y}^{\theta})$ minimization becomes equivalent to minimizing the conditional entropy $\mathbb{H}(p_{X|Y=y}^{\theta})$. Note that the marginal constraint on $Y$ from Equation (2) is verified by construction. To satisfy the marginal constraint on $X$ we consider the $\mathbb{KL}$ divergence. This leads to an alternative definition of the MEC problem with soft constraints, that reads as

**Definition 3.1.** Given random variables $X \in \mathcal{X}$ and $Y \in \mathcal{Y}$, the continuous MEC problem with soft marginal constraints corresponds to the optimization problem

$$\min_{\theta} \mathbb{E}_{y \sim p_Y}\left[\mathbb{H}(p_{X|Y=y}^{\theta})\right] + \lambda_X \mathbb{KL}\left[p_X^{\theta} \parallel p_X\right]. \tag{3}$$

Crucially, we note that the parametric portion of the joint distribution, namely $p_{X|Y}^{\theta}(x|y)$, can be interpreted as a conditional generative model of the variable $X$ given $Y$. As a consequence, the conditional entropy from Definition 3.1, can be interpreted as an expected log-likelihood, leading to

$$\min_{\theta} \mathbb{E}_{x,y \sim p_{X,Y}^{\theta}}\left[-\log\left(p_{X|Y=y}^{\theta}\right)\right] + \lambda_X \mathbb{KL}\left[p_X^{\theta} \parallel p_X\right]. \tag{4}$$

A maximum likelihood solution to the MEC problem in Equation (4) is appealing, because it can be addressed by learning the parameters of an appropriate *conditional* generative model, while approximating the marginal constraints on $X$ through the unconditional version of the model. Nevertheless, this approach bears several challenges:

- Asymmetry: Equation (4) can be used to minimize the conditional entropy $\mathbb{H}(p_{X|Y=y}^{\theta})$. The learned conditional generative model can be used to generate samples from variable $X$ given $Y$, but not vice-versa.

- Marginal constraint: in principle, exact matching requires $\lambda_X \to \infty$, but this choice leads to degenerate solutions to the MEC problem. The marginal constraint from Definition 3.1, despite being *soft*, should strive to keep $p_X^{\theta}$ anchored to $p_X(x)$, which is not known.

To address the first challenge, we introduce a second family of parametric models $p_{Y|X}^{\phi}(y|x)p_X(x)$, this time corresponding to conditional generative model of the variable $Y$ given observations of $X$. Then, we can write a specular version of the MEC problem we defined as

$$\min_{\phi} \mathbb{E}_{x,y \sim p_{X,Y}^{\phi}}\left[-\log\left(p_{Y|X=x}^{\phi}\right)\right] + \lambda_Y \mathbb{KL}\left[p_Y^{\phi} \parallel p_Y\right]. \tag{5}$$

Recall that $p^\theta_{X,Y} = p^\theta_{X|Y} p_Y$ and $p^\phi_{X,Y} = p^\phi_{Y|X} p_X$: it is then reasonable to strive, among all the possible solutions, for $p^\theta_{X,Y} = p^\phi_{X,Y}$. This *joint constraint* can be approximated with a penalty term proportional to the $\mathbb{KL}$ divergence between the two distributions. Interestingly, this coupling allows to implement a practical method that exploits cooperation: we use $p^\phi_{Y|X}$ to improve $p^\theta_{X|Y}$, and vice-versa.

To address the second challenge, and pave the way for our practical implementation, we break the optimization problem by first focusing on respecting the marginal constraints. To do so, we pretrain unconditional models such that $p^{\theta_*}_X(x) \approx p_X(x)$ and $p^{\phi_*}_Y(y) \approx p_Y(y)$. Then, we use $\theta_*$ and $\phi_*$ to initialize the conditional models, and anchor their parameters throughout the optimization such that they do not deviate too much from the pretrained models.

Overall, the method we propose writes as

$$\min_\theta \mathbb{E}_{x,y \sim p^\theta_{X,Y}} \left[ -\log \left( p^\phi_{Y|X=x} \right) \right] + \lambda_X \mathbb{KL} \left[ p^\theta_X \parallel p^{\theta_*}_X \right],$$

$$\min_\phi \mathbb{E}_{x,y \sim p^\phi_{X,Y}} \left[ -\log \left( p^\theta_{X|Y=y} \right) \right] + \lambda_Y \mathbb{KL} \left[ p^\phi_Y \parallel p^{\phi_*}_Y \right],$$

$$(6)$$

where we additionally enforce the approximate joint constraint. Notice the difference with Equation (4) and Equation (5): given the structure of the parametric distributions we use, it is possible to show that $\nabla_\theta \mathbb{E}_{x,y \sim p^\theta_{X,Y}} [-\log p^\theta_{X|Y}] \approx \nabla_\theta \mathbb{E}_{x,y \sim p^\theta_{X,Y}} [-\log p^\phi_{Y|X}]$ whenever $p^\theta_{X,Y} = p^\phi_{X,Y}$ (see Appendix A.1 for more details). Strict adherence to the joint constraint, in principle, allows "swapping" the roles of the conditional models without affecting the optimization dynamics, leading to a cooperative method. In practice, we found trough empirical exploration that such a cooperative formulation (See Appendix A), albeit approximate, proves to be much more stable than the original problem, and consequently decided to adopt it in our implementation, as described next.

### 3.1. Practical implementation

In our implementation, we consider the parametric class of probability distributions associated to denoising diffusion probabilistic models (DDPM) (Sohl-Dickstein et al., 2015; Ho et al., 2020). These models enjoy excellent performance in fitting complex multimodal data, and allow accurate estimation of information metrics (Franzese et al., 2024; Kong et al., 2024; Bounoua et al., 2024b; Dewan et al., 2024).

**DDPM.** These generative models are characterized by a forward process, that is fixed to a Markov chain that gradually adds Gaussian noise to the data according to a carefully selected variance schedule $\beta_t$, i.e. $x_t = \sqrt{1-\beta_t} x_{t-1} +$ $\sqrt{\beta_t} \epsilon$ with $\epsilon \sim \mathcal{N}(0, I)$. Interestingly, an arbitrary portion of this forward chain can be efficiently simulated through the equality in distribution $x_t = \sqrt{\bar\alpha_t} x_0 + (\sqrt{1 - \bar\alpha_t}) \epsilon$, with $x_0 \sim p_X$ and $\alpha_t = 1 - \beta_t$, $\bar\alpha_t = \prod_{s=1}^t \alpha_s$.

The corresponding discrete-time reverse process, that has a Markov structure as well, is used for generative purposes. The model generates data through the iterative sampling process $p^\theta_X(x_{0...T}) = \prod_{t=1}^T p^\theta(x_{t-1}|x_t) p^\theta(x_T)$, where $p^\theta(x_T) = \mathcal{N}(x_T; 0, I)$ and typically $p^\theta(x_{t-1}|x_t)$ is a Gaussian transition kernel with mean $\frac{1}{\sqrt{\alpha_t}} \left( x_t - \frac{\beta_t}{\sqrt{1-\bar\alpha_t}} \epsilon^\theta(x_t, t) \right)$ and covariance $\beta_t I$. Intuitively, starting from a simple distribution $x_T \sim \mathcal{N}(\mathbf{0}, \mathbf{I})$, samples are generated by a denoising network $\epsilon^\theta$, that removes noise over $T$ denoising steps. A simple way to learn the denoising network $\epsilon^\theta$ is to consider a re-weighted variational lower bound of the expected marginal likelihood, where the problem $\arg\min_\theta \mathbb{KL} \left[ p_X \parallel p^\theta_X \right]$ becomes

$$\arg\min_\theta \sum_{t=1}^T \mathbb{E}_{\epsilon \sim \mathcal{N}(0,I), x_0 \sim p_X} \left[ ||\epsilon - \epsilon^\theta(x_t, t)||^2 \right]. \quad (7)$$

This simple formulation has been extended to conditional generation (Ho & Salimans, 2021), whereby a conditioning signal $y$ injects "external information" in the iterative denoising process. This requires a simple extension to the denoising network such that it can accept the conditioning signal: $\epsilon^\theta(x_t, y, t)$. During training, a randomized approach allows to learn both the conditional and unconditional variants of the denoising network, for example by assigning a null value to the conditioning signal, e.g. $y = \emptyset$.

In Equation (6), the log-likelihood emerges as a critical quantity to address the MEC problem. In the ideal conditions of a *perfect* denoising network, the difference between predicted and actual noise can be used, in the limit of infinite number of denoising steps, to compute exactly such quantity (Kong et al., 2023). We use these results to compute the log-likelihoods through Monte Carlo estimation techniques

$$-\log p_\theta(x_0) \approx \text{const} + \frac{1}{2} \sum_{t=1}^T \mathbb{E}_\epsilon \left[ ||\epsilon - \epsilon^\theta(x_t, t)||_2^2 \right], \quad (8)$$

where the unspecified constant does not depend neither on $x_0$ nor on $\theta$, and is consequently irrelevant for optimization purposes. This approach can be trivially generalized to the case of a conditional denoising network $\epsilon^\theta(x_t, y, t)$.

**Our method: DDMEC .** We being by pretraining *unconditional* models such that $p^{\theta_*}_X(x) \approx p_X(x)$ and $p^{\phi_*}_Y(y) \approx$

$p_Y(y)$. Then, we use $\theta_*$ and $\phi_*$ to initialize conditional models $p^\theta_{X|Y}$ and $p^\phi_{Y|X}$, which use denoising networks that accept additional conditioning signals, following Zhang et al. (2023). Next, we interpret the optimization expressed in Equation (6) as a model fine-tuning objective, which is reminiscent of the work by Fan et al. (2023).

$$\min_\theta \mathbb{E}_{x,y \sim p^\theta_{X,Y}} r^\phi(y,x) + \tilde{\lambda}_X \mathbb{KL}\left[p^\theta_X \parallel p^{\theta_*}_X\right],$$

$$\min_\phi \mathbb{E}_{x,y \sim p^\phi_{X,Y}} r^\theta(x,y) + \tilde{\lambda}_Y \mathbb{KL}\left[p^\phi_Y \parallel p^{\phi_*}_Y\right], \quad (9)$$

where $r^\phi = -\log p^\phi_{Y|X}$ and $r^\theta = -\log p^\theta_{X|Y}$ are reward signals striving to minimize the conditional entropies, and $\tilde{\lambda}_X, \tilde{\lambda}_Y$ are scaling factors used for fine-tuning, that no longer require to be extremely large. Furthermore, we enforce the joint constraints via extra penalty terms $\mathbb{KL}\left[p^\theta_{X,Y} \parallel p^\phi_{X,Y}\right], \mathbb{KL}\left[p^\phi_{X,Y} \parallel p^\theta_{X,Y}\right]$.

Fine-tuning DDPMs introduces significant computational overhead. To address this, various studies have explored supervised methods (Lee et al., 2023; Wu et al., 2023) or reinforcement learning. In (Clark et al., 2023; Xu et al., 2024), fine-tuning is achieved through direct back-propagation through the reverse process, which can be costly. Alternative methods use proximal policy optimization (PPO) (Fan et al., 2023; Black et al., 2024; Uehara et al., 2024), leading to improved stability. Note that (Fan et al., 2023) incorporates KL-regularization to maximize the reward signal, while ensuring fidelity to the pretrained model, which is analogous to our soft marginal constraints.

In our implementation, we compute gradients of the reward $\nabla_\theta \mathbb{E}_{p^\theta_{X,Y}} r^\phi(y,x)$ as follows (Fan et al., 2023)

$$\mathbb{E}_{p^\theta_{X,Y}} r^\phi(y,x) \sum_{t=1}^T \nabla_\theta \log p^\theta(x_{t-1}|x_t, y) \quad (10)$$

while the gradient of the marginal constraints $\nabla_\theta \mathbb{KL}\left[p^\theta_X \parallel p^{\theta_*}_X\right]$ are obtained as the approximate gradient of an upper bound (Fan et al., 2023)

$$\sum_{t=1}^T \nabla_\theta \mathbb{E}_{x_t}\left[||\epsilon^\theta(x_t, y, t) - \epsilon^{\theta_*}(x_t, t)||^2\right] \quad (11)$$

Similar expressions apply to the specular model.

Given pretrained models $p^{\theta_*}_X, p^{\phi_*}_Y$, the pseudo-code of our DDMEC method in Algorithm 1 is extremely simple, as it materializes as alternating optimization steps, described (for the top Equation (9)) in Algorithm 2. First, we optimize for the parameters $\theta$ of the model $p^\theta_{X|Y=y}$, while fixing

---

**Algorithm 1** DDMEC  Training Loop

**Input:** $\theta_*, \phi_*$
Initialize $\theta \leftarrow \theta_*, \phi \leftarrow \phi_*$
**repeat**
    Call Algorithm 2 with $y \sim p_Y, \theta, \theta_*, \phi$
    Call Algorithm 2 with $x \sim p_X, \phi, \phi_*, \theta$
**until** Converged

---

**Algorithm 2** DDMEC  Training Step

**Input:** $y, \theta, \theta_*, \phi$
$x \sim p^\theta_{X|Y=y}, t \sim \mathcal{U}[0, T], \epsilon \sim \mathcal{N}(0, I)$
Update $\theta$ using Equations (10) and (11)
Update $\phi$ using $\nabla_\phi \mathbb{E}_{y_t,t}\left[||\epsilon - \epsilon_\phi(y_t, x, t)||^2\right]$

---

the parameters $\phi$ of the specular model $p^\phi_{Y|X}$, which we use as a *reward* term. Then we adapt the parameters $\phi$ to ensure $p^\phi_{X,Y} \approx p^\theta_{X,Y}$: this is achieved by noting that we can adapt Equation (7) to this purpose, whereby the parameters $\theta$ are now fixed. In the second phase (which can described as the specular version of Algorithm 2), we optimize for the parameters $\phi$ of the model $p^\phi_{Y|X=x}$, while fixing the parameters $\theta$ of the model $p^\theta_{X|Y}$ using the corresponding reward term. Finally, in a specular manner to the first phase, we ensure coherency of the two models by adapting $\theta$ such that $p^\theta_{X,Y} \approx p^\phi_{X,Y}$, thus satisfying the joint constraint.

## 4. Experiments

DDMEC  is a general method that can be applied across a variety of data domains, as it relies on an information-theoretic measure to match unpaired entities. Next, we demonstrate DDMEC  versatility using two realistic pairing tasks that use various data modalities, including multi-omics and image data. We compare DDMEC  to state-of-the-art methods for each task, and measure performance using domain-specific metrics. More details about DDMEC  implementation, and our experimental protocol are given in Appendix A.

### 4.1. Multi-omics single-cell alignment

Single-cell measurements techniques, such as mRNA sequencing for whole-transcriptome analysis at the single-cell level (Tang et al., 2009), have been adapted and commercialized by companies which developed platforms to facilitate scalable and efficient single-cell transcriptomics and multi-omics data collection. This data provides a detailed snapshot of the heterogeneous landscape of cells in a sample, and can be used to study the cell developmental trajectories across time, for example. The availability of multi-omics measurements – capturing various properties of a cell, such as gene expression, mRNA transcriptomes, chromatin accessibility,

histone modifications, to name a few – calls for data integration methods to combine a variety of modalities (Xi et al., 2024). Unfortunately, current measurement techniques are destructive: it is hard to obtain multiple types of measurements from the same cell. Furthermore, it is well-known that different cell properties, such as transcriptional and chromatin profiles, cannot be matched using the geometric properties of features in the two domains. Then, pairing single-cell data modalities requires methods that do not rely on either common cells or common features across the data types (Welch et al., 2017; Amodio & Krishnaswamy, 2018; Welch et al., 2019; Stuart et al., 2019).

**Baselines.** We compare our proposed method DDMEC to several baselines from both the machine learning and bioinformatics literature, including SCOT (Demetci et al., 2022), MMD-MA (Liu et al., 2019), UNIONCOM (Cao et al., 2020), and SCTOPOGAN (Singh et al., 2023). SCOT proposes a variant of an OT formulation based on the Gromov-Wasserstein distance, which preserves local neighborhood geometry when transporting data points. MMD-MA is a global manifold alignment algorithm based on the maximum mean discrepancy (MMD) measure. UNIONCOM performs unsupervised topological alignment for single-cell multi-omics data, emphasizing both local and global alignment. SCTOPOGAN uses topological autoencoders to obtain latent representations of each modality separately; a topology-guided Generative Adversarial Network then aligns these latent representations into a common space. We compare our method to INFOOT (Chuang et al., 2023) in Appendix B.1. All alternative methods we consider require a choice of distance or similarity measures, which is a pain point that our method DDMEC completely eliminates.

**Datasets.** We evaluate our method on single-cell multi-omics datasets: the peripheral blood mononuclear cells (PBMC) dataset and the bone marrow (BM) dataset. The PBMC dataset comprises healthy human peripheral blood mononuclear cells, profiled using the 10x Genomics multiome protocol, which enables simultaneous measurement of gene expression (RNA) and chromatin accessibility (ATAC) from the same cells. This dataset includes a total of 11,910 cells, encompassing 7 major immune cell types that are further subdivided into 20 finer-grained cell subclasses. The BM dataset consists of human bone marrow cells profiled using the CITE-seq protocol (Stoeckius et al., 2017), which jointly captures gene expression (RNA) and protein abundance via antibody-derived tags (ADT). A set of 10,235 cells are randomly selected from each modality based on the major cell type labels.

For both datasets, we adopt the data preprocessing and evaluation pipeline described in (Singh et al., 2023), resulting in 50-dimensional embeddings per modality. To assess the

| Method | Celltype Acc ↑ | Subcelltype Acc ↑ |
|---|---|---|
| | PBMC | |
| UnionCom⋆ | 34.8 ± 10.9 | 22.9 ± 7.2 |
| MMD-MA⋆ | 28.3 ± 6.4 | 10.2 ± 4.8 |
| SCOT ⋆ | 12.9 ± 1.1 | 2.4 ± 0.2 |
| scTopoGAN ⋆ | 61.7 ± 8.6 | 41.3 ± 6.5 |
| **DDMEC** | **66.3 ± 2.6** | **46.0 ± 0.5** |
| | BM | |
| UnionCom ⋆ | 51.8 ± 3.7 | 20.9 ± 2.6 |
| MMD-MA⋆ | 38.8 ± 17.9 | 10.4 ± 8.4 |
| SCOT⋆ | **90.5** ± 0.0 | 31.6 ± 0.0 |
| scTopoGAN⋆ | 50.9 ± 14.7 | 22.5 ± 5.4 |
| **DDMEC** | 77.3±0.1 | **44.2±0.1** |

Table 1: Single-Cell alignment experiments

quality of the coupling, we compute cross-modal neighborhood consistency: for each cell in one modality, we identify its $k = 5$ nearest neighbors in the aligned space from the other modality using Euclidean distance. We then evaluate the proportion of cases where the cell's class or subclass label matches the majority label among its neighbors. The resulting metrics are reported as the *Celltype Acc* and *Subcelltype Acc*, respectively. In this experiment, DDMEC is trained once, and inference is conducted five times with different seeds.

**Results.** Section 4.1 presents the quantitative results for the single-cell alignment task. Results marked with ⋆ are reported directly from (Singh et al., 2023). We observe that DDMEC consistently outperforms existing baselines on the PBMC dataset, achieving superior performance in aligning both coarse-grained cell types and fine-grained cell subclasses. On the BM dataset, DDMEC obtains the best performance for subclass-level alignment and ranks second for cell-type alignment.

Notably, DDMEC is the only method that demonstrates robust performance across both datasets, whereas alternative approaches exhibit inconsistent behavior—e.g., SCOT performs well on BM but fails to generalize to PBMC. In contrast to existing methods that learn deterministic one-to-one mappings between modalities, DDMEC is fundamentally generative. It learns to sample from a coupling distribution rather than enforcing a fixed correspondence. To compute alignment metrics, we draw a sample in the target modality conditioned on a source cell, then identify the closest observed cell in the dataset using Euclidean distance. Figure 1 illustrates conditional generation with DDMEC using UMAP projections.

### 4.2. Unpaired image translation

This is a well-known problem in computer vision, where, in the absence of paired data (the joint distribution), the

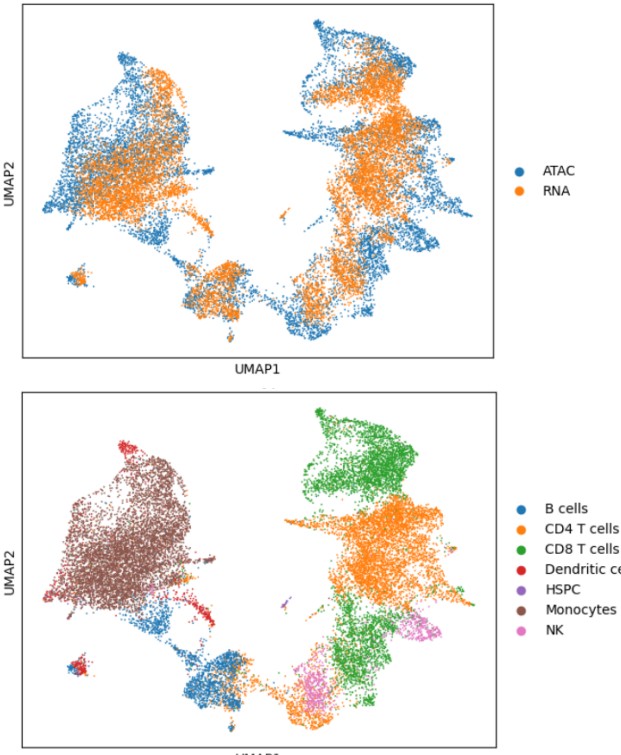

Figure 1: Conditional generation with DDMEC on PBMC dataset. **Top:** UMAP visualizations of source and generated data. **Bottom:** points are colored by cell type, illustrating how well DDMEC preserves cell type separation.

objective is to discover the correct mapping between two image domains. In this work, we show that unpaired image translation can be framed as a MEC problem, where the goal is to learn the correct joint distribution between two unpaired image domains, $\mathcal{X}$ and $\mathcal{Y}$, respectively. Given the growing popularity of diffusion models in image-related tasks, pretrained weights for various image domains are available: we leverage them in our method DDMEC , as done e.g. by Zhang et al. (2023).

**Baselines.** As the literature on image translation is vast, here we primarily focus on the unpaired case, and compare our method to a vast range of alternatives. Generative Adversarial Network (GAN) have been widely applied to this domain (Pang et al., 2021). These methods can be broadly categorized into those focusing on cycle-consistency, which enforces bidirectional mappings between image domains, such as CYCLEGAN (Zhu et al., 2017), DUALGAN (Yi et al., 2017), SCAN (Van Gansbeke et al., 2020), and U-GAT-IT (Kim, 2019); the second category uses distance-based methods, such as DISTANCEGAN (Benaim & Wolf, 2017), GCGAN (Fu et al., 2019), CUT (Park et al., 2020), and LSESIM (Zheng et al., 2021). Diffusion-based models,

which are related to our method, have also been explored for unpaired image translation. UNIT-DDPM (Sasaki et al., 2021) learns two conditional models along with two additional domain translation models, incorporating a GAN-like cycle-consistency loss. ILVR (Choi et al., 2021) and SDEDIT (Meng et al., 2021) utilize a diffusion model in the target domain while conditioning on a source image to refine the sampling procedure for image translation. EGSDE (Zhao et al., 2022) employs an energy function pretrained on both source and target domains to guide the inference process. Similarly, SDDM (Sun et al., 2023) introduces manifold constraints, forcing distributions at adjacent time steps to be decomposable into denoising and refinement components. Compared to these methods, DDMEC leverages two conditional models, one per domain, which can be initialized using pretrained unconditional diffusion models. By design, our method does not require comparable domains and does not rely on a specific image similarity measure. We report results for two values of the guidance coefficient, a parameter influencing conditional generation.

**Datasets.** We adopt the same experimental validation protocol as described by Zhao et al. (2022), where all images are resized to a resolution of $256 \times 256$. We use the AFHQ (Choi et al., 2020) dataset, consisting of high-resolution animal face images across three domains: CAT, DOG, and WILD. This dataset exhibits relatively large variations within and between domains, with 500 test images per domain. We compute the performance of our method DDMEC and compare it to the baselines on CAT→DOG and WILD→DOG tasks. Furthermore, we employ the CELEBA-HQ (Karras, 2017) dataset, which comprises high-resolution human facial images categorized into two distinct domains: MALE and FEMALE. To evaluate the efficacy of our proposed approach relative to existing baselines, we conduct experiments on the domain translation task from MALE to FEMALE.

In addition, due to the bidirectional architecture of DDMEC - which leverages two conditional generative models - our framework also inherently supports translation in the reverse direction. The corresponding results for the FEMALE → MALE task are presented in Appendix B.

**Results.** In Table 2, we present the quantitative results of DDMEC : results for alternative methods, marked with ⋆, are reported as obtained in (Sun et al., 2023; Zhao et al., 2022). DDMEC results are reported using 5 seeds. The evaluation is based on generation quality, measured by the Fréchet Inception Distance (FID) score (Heusel et al., 2017) (lower is better), and the fidelity to the source domain, assessed using structural Similarity Index Measure (SSIM) score (Wang et al., 2004) (higher is better). Note that quality and fidelity can be thought of as divergent objectives: high

quality does not imply high fidelity and vice-versa.

On the AFHQ dataset, GAN-based methods generally suffer from low image quality, except for STARGAN, which achieves a low FID but performs poorly on SSIM. In contrast, diffusion-based methods demonstrate superior performance compared to GAN-based approaches. DDMEC achieves the best FID score in the CAT→DOG task and the highest SSIM in the WILD→DOG task while maintaining comparable results on the remaining metrics. Overall, DDMEC strikes the best balance between high-quality image generation and accurate alignment with the target domain.

Our results on the CELEBA-HQ dataset, demonstrate that DDMEC outperforms competitors on both FID and SSIM even with only 50 sampling steps. Specifically, at 50 steps the FID improves by approximately 1 point and the SSIM by 0.02 points, with an even greater improvement (a 3 point FID reduction) when using 100 sampling steps: it is well-known in the generative modeling literature that these improvements are significant. With the CELEBA-HQ dataset, DDMEC benefits from a larger training set than in AFHQ animal dataset, and achieves state-of-the-art performance on image translation.

Table 2: Quantitative image translation results.

| Model | FID↓ | SSIM↑ |
|---|---|---|
| CAT→DOG | | |
| CycleGAN⋆ | 85.9 | - |
| MUNIT⋆ | 104.4 | - |
| DRIT⋆ | 123.4 | - |
| Distance⋆ | 155.3 | - |
| SelfDistance⋆ | 144.4 | - |
| GCGAN⋆ | 96.6 | - |
| LSeSim⋆ | 72.8 | - |
| ITTR (CUT)⋆ | 68.6 | - |
| StarGAN v2⋆ | **54.88 ± 1.01** | 0.27 ± 0.003 |
| CUT⋆ | 76.21 | **0.601** |
| SDEdit⋆ | 74.17 ± 1.01 | **0.423 ± 0.001** |
| ILVR⋆ | 74.37 ± 1.55 | 0.363 ± 0.001 |
| EGSDE⋆ | 65.82 ± 0.77 | 0.415 ± 0.001 |
| SDDM⋆ | 62.29 ± 0.63 | **0.422± 0.001** |
| 50 Sampling steps | | |
| **DDMEC** (guidance=9) | **60.70 ± 1.07** | 0.410 ± 0.001 |
| **DDMEC** (guidance=7) | **58.50 ± 0.96** | 0.404 ± 0.001 |
| 100 Sampling steps | | |
| **DDMEC** (guidance=9) | **60.51 ± 1.01** | 0.403 ± 0.001 |
| **DDMEC** (guidance=7) | **57.89 ± 0.37** | 0.397 ± 0.001 |
| WILD→DOG | | |
| SDEdit⋆ | 68.51 ± 0.65 | 0.343 ± 0.001 |
| ILVR⋆ | 75.33 ± 1.22 | 0.287 ± 0.001 |
| EGSDE⋆ | 59.75 ± 0.62 | 0.343 ± 0.001 |
| SDDM⋆ | **57.38 ± 0.53** | 0.328 ± 0.001 |
| 50 Sampling steps | | |
| **DDMEC** (guidance=9) | 62.03 ± 1.18 | **0.360 ± 0.002** |
| **DDMEC** (guidance=7) | 60.67 ± 1.01 | **0.353 ± 0.004** |
| 100 Sampling steps | | |
| **DDMEC** (guidance=9) | 62.09 ± 0.59 | **0.356± 0.001** |
| **DDMEC** (guidance=7) | 59.22 ± 0.35 | **0.346 ± 0.001** |
| MALE→FEMALE | | |
| SDEdit⋆ | 49.43 ± 0.47 | 0.572 ± 0.000 |
| ILVR⋆ | 46.12 ± 0.33 | 0.510 ± 0.001 |
| EGSDE⋆ | 41.93 ± 0.11 | 0.574 ± 0.000 |
| SDDM⋆ | 44.37± 0.23 | 0.526 ± 0.001 |
| 50 Sampling steps | | |
| **DDMEC** (guidance=2.5) | **40.73 ±0.61** | **0.593 ± 0.003** |
| **DDMEC** (guidance=2) | **36.99± 0.83** | 0.556 ± 0.002 |
| 100 Sampling steps | | |
| **DDMEC** (guidance=2.5) | **38.93 ± 0.37** | **0.588 ± 0.002** |
| **DDMEC** (guidance=2) | **34.86 ± 0.70** | 0.549 ± 0.002 |

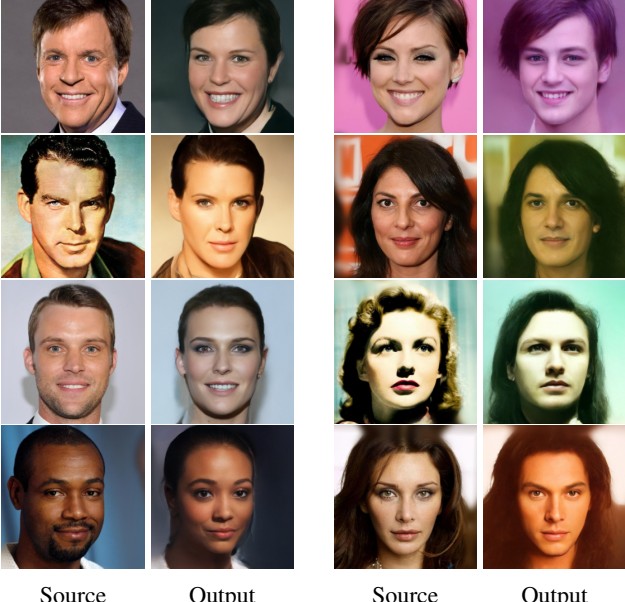

Source     Output     Source     Output

Figure 2: Qualitative results of DDMEC -100 (guidance=2.5) on CELEBA-HQ. **Left:** :MALE→FEMALE and **Right:** FEMALE→MALE . Source domain image is used to generate the target female image.

This outcome aligns with expectations, as DDMEC is designed to reduce uncertainty and enforce adherence to marginal constraints. In Figures 2 and 3, qualitative results further confirm the performance of DDMEC in this task. Additional results are available in Figures 6 and 9.

We investigate the effect of the guidance scale which acts as a temperature like parameter that controls conditioning strength—in our diffusion-based unpaired image translation

framework. A test-time ablation study (Figure 8a) reveals a trade-off: higher guidance improves SSIM (structural similarity) but degrades FID (image realism), while lower guidance yields the opposite. This illustrates how guidance balances modality fidelity and mutual information maximization under the MEC framework.

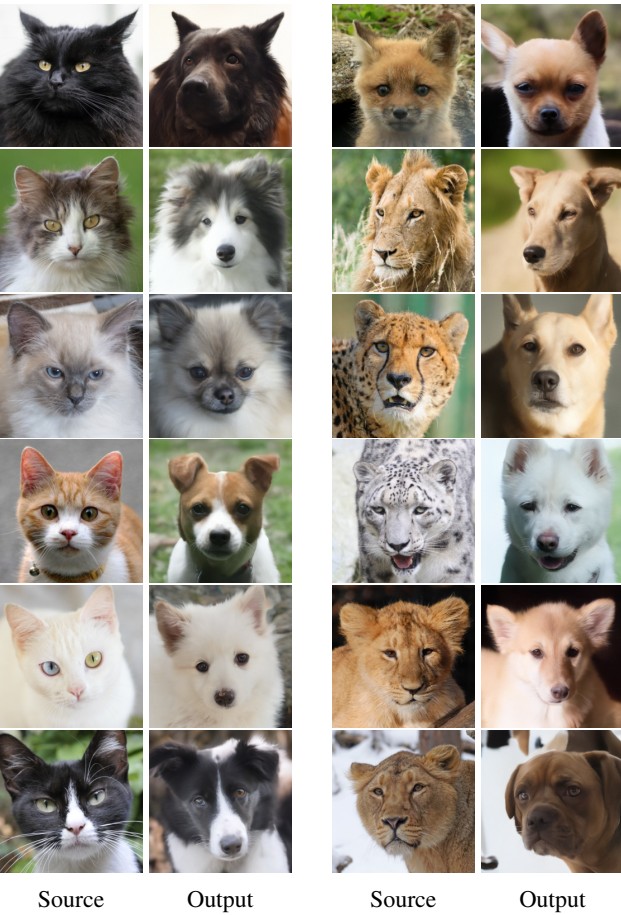

|          | Source | Output | Source | Output |

Figure 3: DDMEC (guidance=7) CAT→DOG (*Left*) and WILD→DOG image (*right*) translation examples. Source domain image is used to generate the target dog image.

## 5. Conclusion

The machine learning community has recently directed substantial effort toward designing multimodal models, as they reflect the inherently multi-faceted nature of the real world. These models often achieve superior performance on downstream tasks compared to unimodal counterparts. However, the intrinsic complexity of multimodal data introduces significant challenges. In this work, we addressed the critical problem of coupling data represented by diverse modalities. The coupling problem has been widely studied in the literature, often framed as an optimal transport problem or approached with specialized architectures tailored to spe-

cific domains, such as images or language. However, existing methods typically rely on geometric spaces to compute costs, mappings, and similarities between data points.

We proposed a novel method that shifts the focus toward information and uncertainty quantification, thereby circumventing the limiting assumptions of prior approaches. Specifically, we studied the coupling problem through the lens of minimum entropy coupling. Since prior work on MEC has largely been confined to discrete distributions, we extended this framework to continuous distributions. Our key idea lies in introducing a parametric class of joint distributions reinterpreted as conditional generative models, augmented with terms to enforce adherence to marginal constraints. Our approach uses two models, which alternately optimize their objectives while approximately satisfying marginal constraints.

The resulting method enables sampling and generation in either direction between modalities, without requiring specialized embeddings or strict geometric assumptions. Furthermore, it is adaptable to complex settings beyond one-to-one matching between modalities. We validated the performance of our approach in two domains. First, we applied it to multi-omics sequencing data, and we compared our method against several state-of-the-art alternatives that rely on predefined measures for data comparison and coupling cost definition. Our approach, being more general and free from stringent assumptions, achieves performance on par with or superior to these alternatives. Second, we evaluated our method in the image translation domain, comparing it to a range of approaches from the literature. Our method demonstrated superior performance across widely recognized metrics for image quality and coherence, by striking a good balance between these often conflicting measures.

## Acknowledgment

Pietro Michiardi was partially funded by project MUSE-COM$^2$ - AI-enabled MUltimodal SEmantic COMmunications and COMputing, in the Machine Learning-based Communication Systems, towards Wireless AI (WAI), Call 2022, ChistERA.

## Impact Statement

This paper aims to advance Machine Learning by improving unpaired data translation, with potential benefits for applications such as multi-omics alignment in biology and image translation. However, these advancements may also increase risks associated with generative models, including the generation of deepfakes. Responsible and ethical use is advised.

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

# A. Additional Details

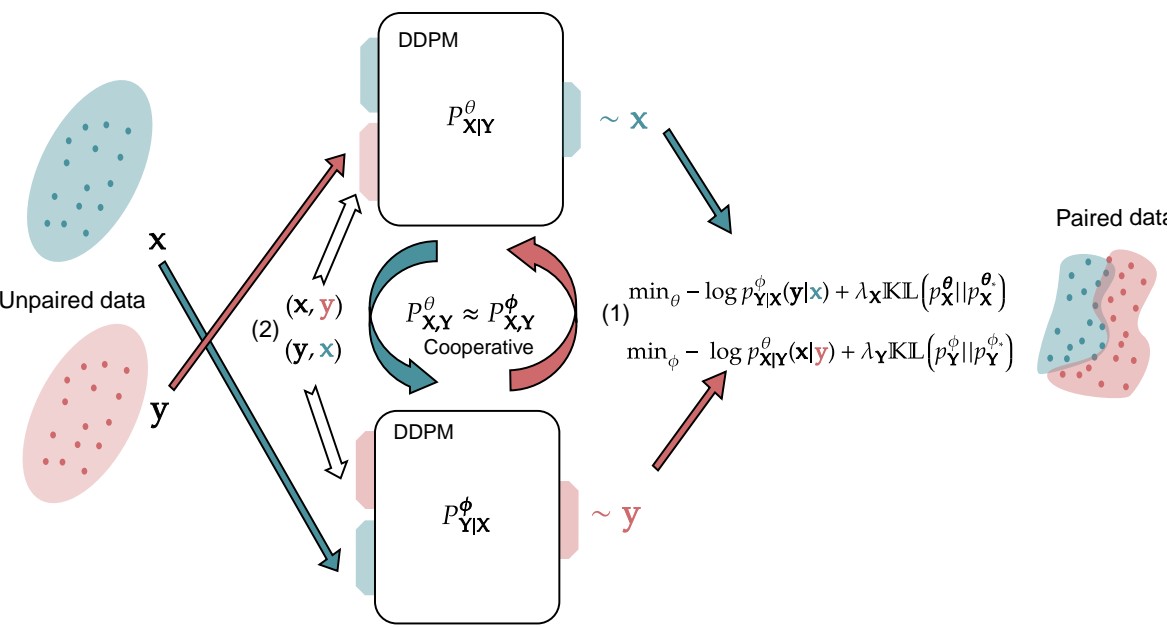

Figure 4: Overview of the DDMEC methodology. The phase (1) corresponds to the procedure described in Algorithm 2, Lines 2 and 2, and involves generating samples (depicted in red and blue) conditioned on inputs $x$ and $y$ drawn from their respective marginals. These samples are then used to evaluate the loss defined in Equation (9), which is practically optimized using PPO (Fan et al., 2023). The phase (2) corresponds to Line 2 of Algorithm 2, wherein the joint consistency constraint is enforced by updating the model with the previously generated sample pairs. Both phases require coordination between the two models, alternating their roles as outlined in Algorithm 1.

## A.1. Details on Swapping the Roles of the Conditional Models

In this part we provide more details about the justification of :

$$\nabla_\theta \mathbb{E}_{x,y \sim p^\theta_{X,Y}}[-\log p^\theta_{X|Y}] \approx \nabla_\theta \mathbb{E}_{x,y \sim p^\theta_{X,Y}}[-\log p^\phi_{Y|Y}] \tag{12}$$

*whenever* $p^\theta_{X,Y} = p^\phi_{X,Y}$.

Equation (12) reads:

$$\nabla_\theta \int p_Y(y)\, p^\theta_{X|Y}(x|y) \log p^\theta_{X|Y}(x|y)\, dx\, dy. \tag{13}$$

Moving the gradient $\nabla_\theta$ inside the integral and applying the chain rule, we obtain:

$$\int p_Y(y)\, \nabla_\theta \left( p^\theta_{X|Y}(x|y) \right) \log p^\theta_{X|Y}(x|y)\, dx\, dy$$
$$+ \int p_Y(y)\, p^\theta_{X|Y}(x|y)\, \nabla_\theta \left( \log p^\theta_{X|Y}(x|y) \right)\, dx\, dy. \tag{14}$$

The second term simplifies to zero:

$$\int p_Y(y)\, p^\theta_{X|Y}(x|y)\, \nabla_\theta \left(\log p^\theta_{X|Y}(x|y)\right) dx\, dy = \int p_Y(y)\, \nabla_\theta \left(p^\theta_{X|Y}(x|y)\right) dx\, dy$$

$$= \nabla_\theta \int p_Y(y)\, p^\theta_{X|Y}(x|y)\, dx\, dy$$

$$= \nabla_\theta 1 = 0. \tag{15}$$

Assuming $p^\theta_{X,Y} = p^\phi_{X,Y}$, $p^\theta_X = p_X$, and $p^\phi_Y = p_Y$, the first term rewrites as:

$$\int p_Y(y)\, \nabla_\theta \left(p^\theta_{X|Y}(x|y)\right) \log \frac{p^\phi_{Y|X}(y|x)p_X(x)}{p_Y(y)}\, dx\, dy$$

$$= \int p_Y(y)\, \nabla_\theta \left(p^\theta_{X|Y}(x|y)\right) \left(\log p^\phi_{Y|X}(y|x) + \log p_X(x) - \log p_Y(y)\right) dx\, dy$$

$$= \nabla_\theta \int p_Y(y)\, p^\theta_{X|Y}(x|y) \log p^\phi_{Y|X}(y|x)\, dx\, dy, \tag{16}$$

which corresponds to the right-hand side (r.h.s) of Equation (12).

Indeed, the additional terms vanish:

$$\int p_Y(y)\, \nabla_\theta \left(p^\theta_{X|Y}(x|y)\right) \log p_X(x)\, dx\, dy = \int \nabla_\theta \left(p^\theta_X(x)\right) \log p_X(x)\, dx$$

$$= \int \nabla_\theta \left(p_X(x)\right) \log p_X(x)\, dx = 0, \tag{17}$$

and similarly,

$$-\int p_Y(y)\, \nabla_\theta \left(p^\theta_{X|Y}(x|y)\right) \log p_Y(y)\, dx\, dy = 0. \tag{18}$$

### A.2. Diffusion Models Training with Reinforcement Learning

Our methodology begins by training unconditional diffusion models for both data modalities. We then use a reinforcement learning technique to train two conditional models (initialized from the first step) in a cooperative manner, allowing them to learn from each other to optimize the joint coupling under MEC constraints and objectives. We formulate this second phase as training diffusion models with reinforcement learning and $\mathbb{KL}$-regularization. We follow the training scheme presented by Fan et al. (2023), where samples are generated conditionally using classifier guidance (Ho & Salimans, 2021) with a DDIM sampler (Song et al., 2020). The generated trajectories are then used to update the diffusion model, which is framed as a Markov Decision Process (MDP), using a policy gradient RL algorithm.

**Reward Estimation** In DDMEC , the reward signals are log-likelihood values mutually generated by the two conditional models. Accurately estimating this signal is crucial for steering training towards the optimal MEC solution. To achieve this, we use multiple Monte Carlo steps to estimate Equation 8.

**Policy Gradient Training** We follow the training procedure of Fan et al. (2023), where, at each step, a batch of samples is generated using DDIM (Song et al., 2020). These generated trajectories are then used to perform multiple gradient updates. Additionally, we apply importance sampling and ratio clipping (Schulman et al., 2017) to improve training stability.

**Classifier-Free Guidance** We employ classifier-free guidance (Ho & Salimans, 2021) in all experiments. This technique enables conditional sampling in step 2. The denoising loss in 2 is optimized to account for the guidance mechanism by randomly dropping 10% of the conditional signal, thereby stabilizing the unconditional model.

### A.3. Technical Details and Hyperparameters

The source [1] is publicly available.

---

[1] https://github.com/MustaphaBounoua/ddmec

**Single-Cell Alignment**    For the PBMC and BM datasets, we utilize the preprocessed versions provided in the official code repository of Singh et al. (2023)[2], along with the accompanying evaluation protocols. Prior to training, we normalize the data by subtracting the mean and scaling to unit variance, while applying outlier mitigation. The model is first trained unconditionally using a DDPM for 100,000 steps with $T = 1000$ diffusion steps. To stabilize training, we additionally incorporate a nearest neighbor retrieval step after each generation, where generated samples are projected back to the closest real data points using Euclidean distance. Subsequently, we train the two conditional models, following the procedure outlined in Algorithm 1. For each training step, we use a batch size of 256 and perform four gradient updates corresponding to line 2, followed by four updates for Line 2. We find it beneficial to accumulate generated samples during training and reuse them in optimizing Line 2. We use a simple MLP network with skip connections and use the Adam optimizer (Kingma, 2014) with a learning rate of $1 \times 10^{-4}$. The KL divergence regularization weight is set to $\lambda = 0.01$ for PBMC and $\lambda = 0.02$ for BM.

**Unpaired Image Translation**    - CAT→DOG and WILD→DOG Tasks: We utilize the pre-trained model from the official implementation of Choi et al. (2021) (`https://github.com/jychoi118/ilvr_adm`) to initialize the dog modality conditional model. For the other domains (CAT, WILD): We train a diffusion model from scratch using the same architecture and hyper-parameters as done in the target domain. - **MALE→FEMALE Task:** We use the publicly available pre-trained model by Zhao et al. (2022) (`https://github.com/ML-GSAI/EGSDE`) for the Female modality. For the Male modality, we train a diffusion model from scratch using the same architecture and hyper-parameters as done in the target domain.

To introduce additional conditioning into the pre-trained diffusion model, we follow the work in (Zhang et al., 2023), where the encoder part of the U-NET is duplicated and used as a conditional encoder. The various hyperparameters are summarized in Table 3. We follow the evaluation protocol described in (Zhao et al., 2022).

| **General Settings** | Dataset | |
| --- | --- | --- |
| | AFHQ | CelebA-HQ |
| Batch Size | 16 | 16 |
| Learning Rate: | $2e-5$ | $2e-5$ |
| Optimizer | ADAM | ADAM |
| Training Steps | 2000 | |
| Weight Decay | 0.0 | |
| **Diffusion Model** | | |
| Noise Scheduler | Linear | Linear |
| Number of Diffusion Timesteps ($T$) | 1000 | 1000 |
| Sampler | DDIM | DDIM |
| Guidance Scale (training) | 7.0 | 7.0 |
| Sampling steps | 50 | 50 |
| Exponential moving average | Yes | Yes |
| **Reinforcement Learning** | | |
| Reward (Monte Carlo steps) | 3 | 3 |
| Policy Update Steps | 4 | 4 |
| Importance Sampling Clipping | $1e-4$ | $1e-4$ |
| $\lambda_1, \lambda_2$ | $1e-3$ | $4e-3$ |
| Gradient Accumulation | 12 | 12 |
| Gradient Clipping | 1.0 | 1.0 |

Table 3: Hyperparameters used for training.

---

[2]`https://github.com/AkashCiel/scTopoGAN`

# B. Additional Results

## B.1. SNARE-seq additional experiments

In this experiment we use the SNARESEQ (Chen et al., 2019) dataset, which links chromatin accessibility with gene expression data on a mixture of four cells types. We use the same preprocessing procedures detailed in (Demetci et al., 2022), which deal with filtering spurious data affected by technical errors, and normalization. Data samples have 1–1 correspondence information, which constitute the groud-truth information used for our performance evaluation. We use the average "fraction of samples closer than the true match (FOSCTTM)" metric introduced by Liu et al. (2019): given a sample in one domain, this amounts to compute the fraction of samples that are positioned more closely to it than its true match after pairing. Results report the average FOSCTTM across all samples, where lower values indicate better performance. We also report the label transfer accuracy as done by Cao et al. (2020), which measures how well sample labels are transferred between domains based on neighborhood alignment. A $k$-nearest neighbor classifier is trained on one domain and used to predict labels in the other. In this experiment, DDMEC is trained once, and inference is conducted five times with different seeds. the FOSCTTM metric, and is on-par with the best method in terms of accuracy. In this experiment, DDMEC is trained once, and inference is conducted five times with different seeds. Unlike other methods, DDMEC is conceptually different as it generates samples rather than learning a deterministic, 1-1 mapping. To compute the different metrics, given a sample from one modality, we use DDMEC to generate a coupling to the other modality, then select the nearest sample from the dataset based on Euclidean distance.

|            | SNAREseq        |         |
|------------|-----------------|---------|
|            | FOS $\downarrow$ | Acc $\uparrow$ |
| UnionCom⋆  | 0.265           | 42.3    |
| MMD-MA⋆    | 0.150           | 94.2    |
| SCOT⋆      | 0.150           | 98.2    |
| InfoOT⋆    | 0.156           | **98.8** |
| **DDMEC**  | **0.147**       | 98.6    |

Table 4: Performance results on **SNAREseq** dataset

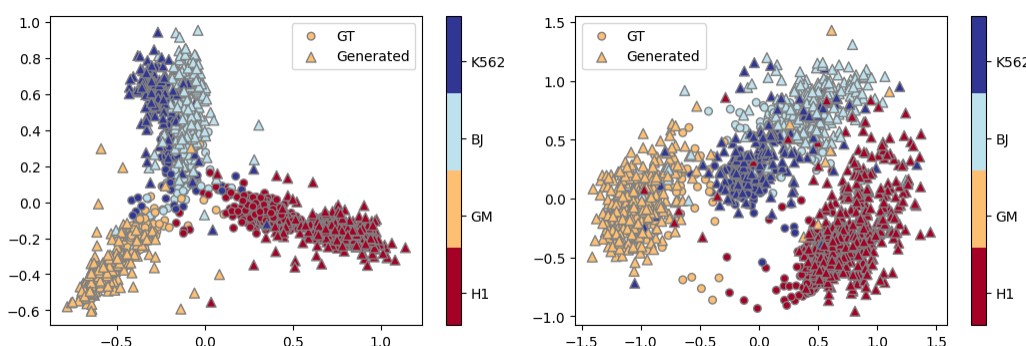

Figure 5: Conditional generation using DDMEC on the SNAREseq dataset. The cell types are indicated by colors. **Top:** generation of chromatin accessibility data using gene expression, **Bottom:** generation of gene expression using chromatin accessibility data.

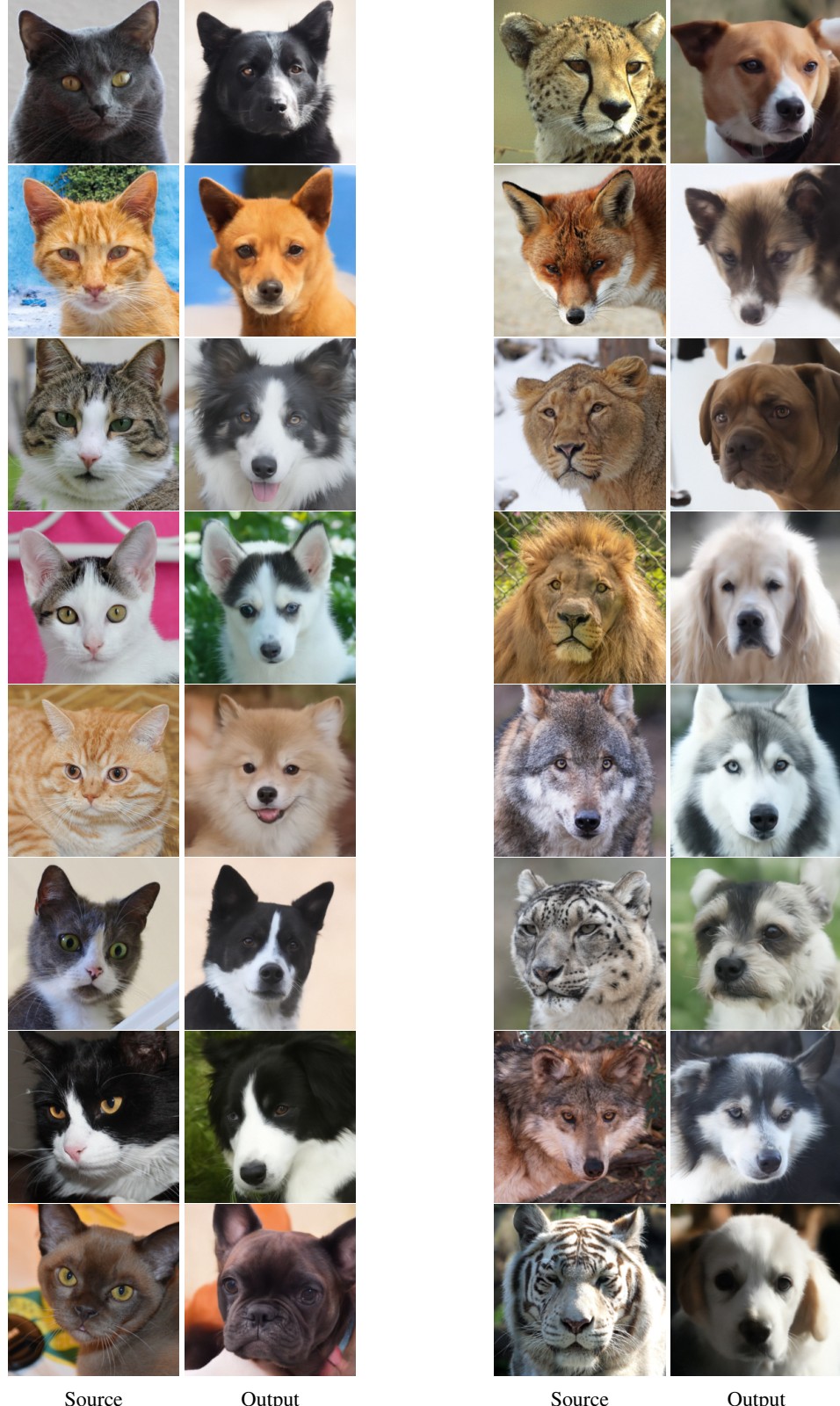

Source      Output          Source      Output

Figure 6: DDMEC (guidance=7) CAT→DOG (*Left*) and WILD→DOG image (*right*) translation examples. Source domain image is used to generate the target dog image.

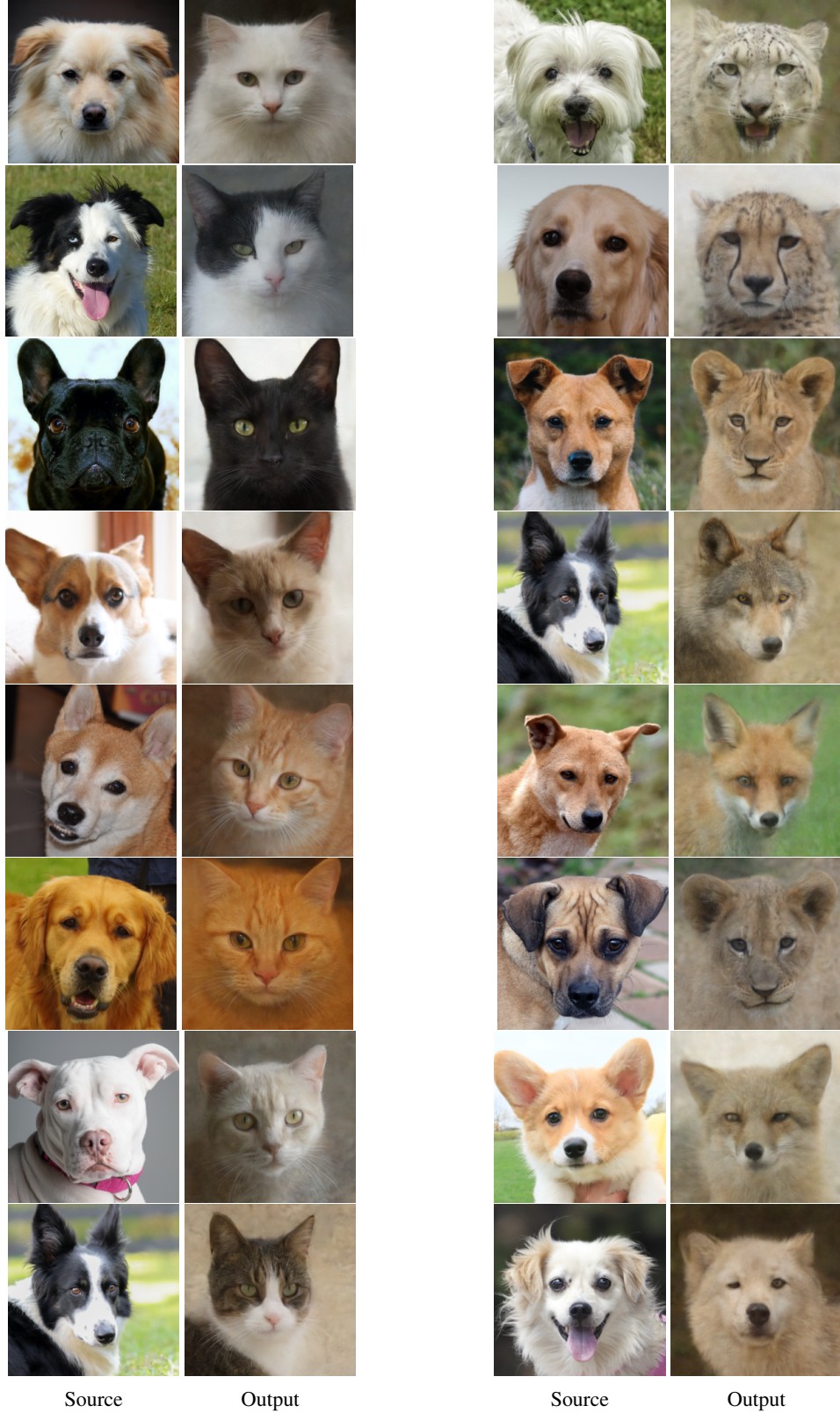

Source      Output             Source      Output

Figure 7: DDMEC (guidance=7) DOG→CAT (*Left*) and DOG→WILD image (*right*) translation examples. Source domain image is used to generate the target Cat/Wild image.

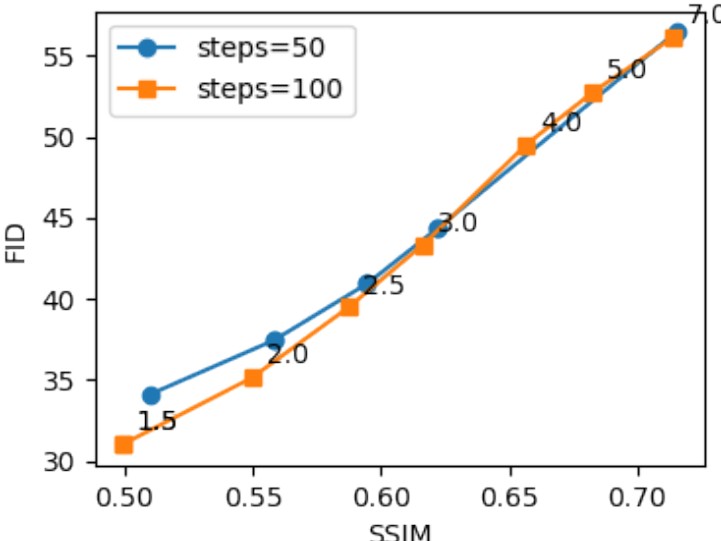

(a) In the guidance scale ablation study, we report the FID and SSIM as a function of the guidance scale on the CelebA-HQ dataset. We notice that an increase in the guidance scale results in more information transfer between the two modalities, leading to a worse FID, and vice versa.

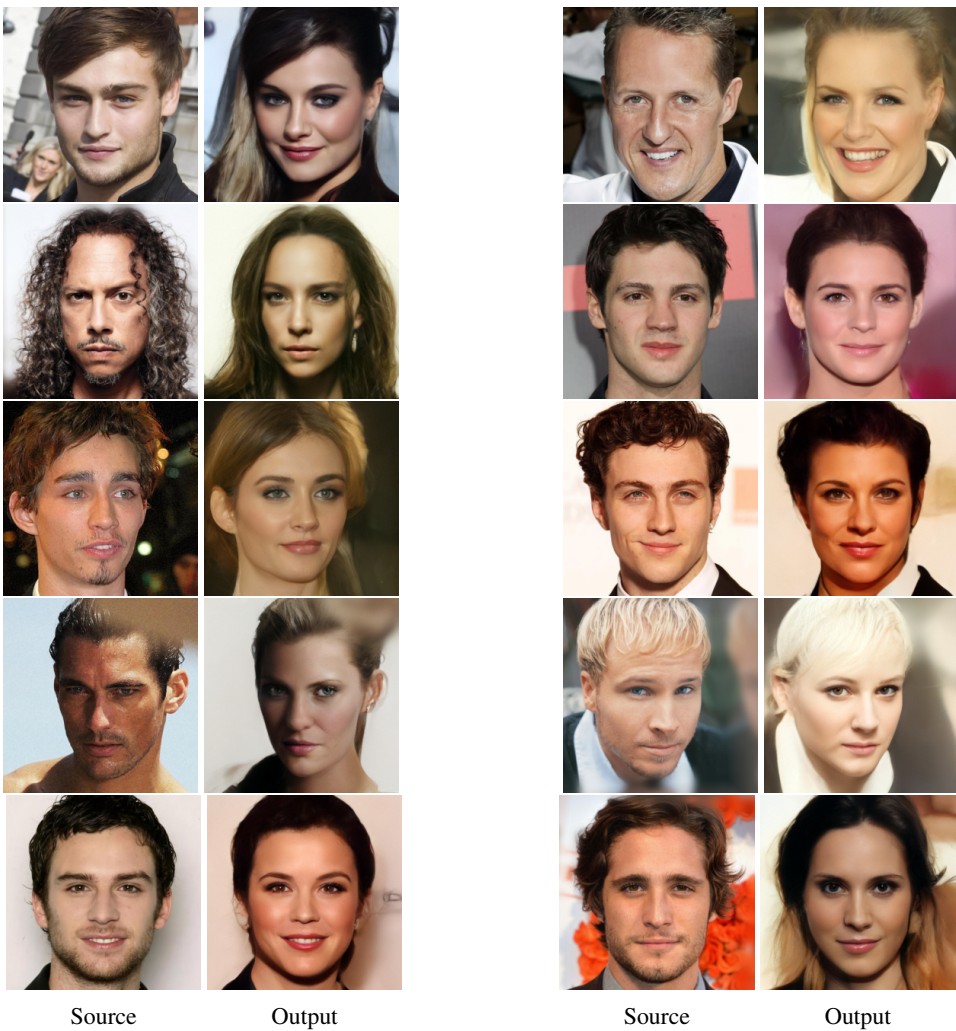

Source Output               Source Output

Figure 9: DDMEC (guidance=2.5) MALE→FEMALE translation examples. Source domain image is used to generate the target female image.

