# OpenReview forum: "Learning to Match Unpaired Data with Minimum Entropy Coupling"
_ICML.cc/2025/Conference — ICML 2025 poster_

### Official Review · Reviewer_2cNi · 2025-03-04

**Overall Recommendation:** 3

**Summary:**

This paper proposes a novel method to solve the continuous Minimum Entropy Coupling (MEC) problem. Specifically, it incorporates generative diffusion models to learn the joint distribution with the minimum joint entropy, while enforcing a relaxed version of the marginal constraints.

**Claims And Evidence:**

In general, the claims made in the paper are supported by clear and convincing evidence. However, I believe the experiments are not extensive enough, and require additional datasets/applications.

**Essential References Not Discussed:**

None

**Experimental Designs Or Analyses:**

I’ve checked the soundness of the experimental designs, and I don’t see any issues.

**Methods And Evaluation Criteria:**

yes

**Other Comments Or Suggestions:**

I believe there’s a typo in line 190 (should be Y|X)

**Other Strengths And Weaknesses:**

Strengths:
1.	The paper considers an important topic in multimodal learning – learning from naturally unpaired data.
2.	The paper is generally clear and easy to follow

Weaknesses:
See questions.

**Questions For Authors:**

Question For Authors
1.	The evaluation of each of the two applications is done using only a single dataset. I find it not convincing enough. Could you add one dataset to each application?
2.	The applications shown in the paper are existent, and the proposed method only marginally improve existing results (e.g., InfoOT, SDDM). As DDMEC is general, could you present another application of it? Please support this with empirical evaluation.
3.	I’m afraid about the data-scalability of DDMEC. Could you elaborate about it? How is it compared to other translation methods such as CycleGAN and SDDM? And compared to InfoOT?

**Relation To Broader Scientific Literature:**

The key contribution of the paper is related to the broader scientific literature through interesting ideas (e.g., combining diffusion models with MEC).

**Theoretical Claims:**

N/A

---

> ### Author Rebuttal · Authors · 2025-03-31
>
> We appreciate the reviewer's insightful feedback and suggestions. Below, we provide responses and additional experiments to address their concern.
>
>  > 1. Could you add one dataset to each application?
>
> - **Additional Image translation experiment**:
>
> We consider the CelebA-HQ dataset (Karras et al., 2017), which features high-quality human face images divided into two domains: Male and Female. Each domain includes approximately 10,000 and 17000 training samples for the male and female modalities. We compare the performance of DDMEC against several baseline methods on the Male-to-Female translation task, following the same evaluation protocol as in (Zhao et al. 2022).
>
> | **Model**                     | **FID ↓**       | **SSIM ↑**       |
> |-------------------------------|-----------------|------------------|
> | **CelebA-HQ (Male → Female)** |                 |                  |
> | SDEdit*                       | 49.43 ± 0.47    | 0.572 ± 0.000    |
> | ILVR*                         | 46.12 ± 0.33    | 0.510 ± 0.001    |
> | EGSDE*                        | 41.93 ± 0.11    | 0.574 ± 0.000    |
> | SDDM*                         | 44.37 ± 0.23    | 0.526 ± 0.001    |
> | **50 sampling steps**         |                 |                  |
> | DDMEC (guidance=2.5)          | **40.73 ± 0.61**| **0.593 ± 0.003**|
> | **100 sampling steps**        |                 |                  |
> | DDMEC (guidance=2.5)          | **38.93 ± 0.37**| **0.588 ± 0.002**    |
>
>
> In our experiment, we follow the same training procedure as for the AFHQ dataset available in the main paper. Our new results, demonstrate that DDMEC outperforms competitors on both FID and SSIM even with only 50 sampling steps. Specifically, at 50 steps the FID improves by approximately 1 point and the SSIM by 0.02 points, with an even greater improvement (a 3-point FID reduction) when using 100 sampling steps: it is well-known in the generative modeling literature that these improvements are significant. With the CelebA-HQ dataset, DDMEC benefits from a larger training set (in the AFHQ animal dataset, each modality has approximately 3000 images) and achieves state-of-the-art performance on image translation. Qualitative results supporting the quantitative evaluations can be viewed via ( https://anonymous.4open.science/api/repo/icml2025_ddmec-7798/file/celeba/qualitative.png?v=7d6cc987 ).
>
> - **Additional Single-Cell data experiment:**
>
> We performed new experiments on a more complex single-cell data alignment task. Since the SNARE dataset we used in the submitted paper is relatively small and simple, our goal is to further substantiate the superiority of DDMEC using the peripheral blood mononuclear cells (PBMC) dataset: this dataset consists of healthy human PBMCs with simultaneous profiling of gene expression (RNA) and chromatin accessibility (ATAC). PBMC contains 11,910 cells, spanning 7 major cell classes that are further divided into 20 cell subclasses. We use the data processing and evaluation pipeline from (Singh, 2023), which results in a 50-dimensional representation for each modality.
>
>
>
> The obtained results available here (https://anonymous.4open.science/r/icml2025_ddmec-7798/PBMC/table.png )  demonstrate that DDMEC performs extremely well also in a high-dimensionality scenario, whereas OT-based methods completely fail. Although OT-based methods are relatively lightweight, they do not scale, motivating the need for more advanced methodologies such as DDMEC. Our method outperforms the best competitor, scTopoGAN, scoring superior performance in terms of both cell type matching rate and subcell matching rate.
>
> > 2. The applications shown in the paper are existent, and the proposed method only marginally improve existing results (e.g., InfoOT, SDDM). As DDMEC is general, could you present another application of it?
>
> The experiments (presented in Q1) include results on the extra CelebA-HQ dataset and an additional PBMC single-cell dataset, both of which are more elaborate and sophisticated. The results demonstrate that DDMEC outperforms and improves upon the state-of-the-art. The reviewer’s suggestion about considering additional applications is both valid and insightful. As suggested by reviewer Acbm, it is possible to consider image-text pairs or multilingual text alignment. However, due to rebuttal  time constraints, we leave this to the camera-ready version.
>
> > 3. data-scalability of DDMEC?
>
> We discuss the scalability of our method against the baselines in our answer to the question 3  of Reviewer   zrWS.
>
> > typo in line 190
>
> We apologize, indeed it should be $Y|X$.
>
> -Singh, et. al, scTopoGAN: unsupervised manifold alignment of single-cell data. Bioinformatics Advances 2023.

---

> > ### Comment · Reviewer_2cNi · 2025-04-06
> >
> > Thank you for the detailed response and the additional experimental results. I appreciate the authors' effort in addressing my concerns. Based on the revisions, I am increasing my score to 3.

---

> > > ### Author Response · Authors · 2025-04-09
> > >
> > > Thank you very much for your valuable suggestions. We’re glad that the additional experiment addressed your concerns and that you updated your evaluation score accordingly.
> > >
> > > Thanks again,
> > > The Authors

---

### Official Review · Reviewer_Acbm · 2025-03-11

**Overall Recommendation:** 4

**Summary:**

This paper proposes minimum entropy coupling (MEC) to align unpaired multimodal data. MEC seeks a joint distribution with the desired marginals that is optimal in the sense of minimum entropy, in comparison to optimal transport approaches which minimize an integrated cost. This sidesteps the difficulty of specifying a cross-modal cost function which in previous work was either achieved using auxiliary labels or with Gromov-Wasserstein OT. To solve MEC over continuous spaces, the joint is factorized into known marginals and unknown conditionals, for which minimum entropy becomes maximum conditional likelihood. This is done for both possible factorizations and the conditional models are trained as denoising diffusion. Experiments are performed on single-cell and image translation datasets.

**Claims And Evidence:**

Yes, I think the claims made in the paper are supported by evidence.

**Essential References Not Discussed:**

The authors do a good job at reviewing the relevant literature. There are a few methods that bring both modalities into a shared latent space and then do OT alignment on that space, e.g., MatchCLOT (Gossi et al., 2023) or propensity score alignment (Xi et al., 2024), but this usually requires some sort of supervision (ground truth pairs, or shared labels) to first learn the latent space so it may not be as relevant here. One simple baseline (though I don't expect this to do well unsupervised) might be to just do separate dimension reduction on both modalities and align them with OT.

Gossi et al., Matching single cells across modalities with contrastive learning and optimal transport, Briefing in Bioinformatics, 2023.

Xi et al., Propensity score alignment of unpaired multimodal data, NeurIPS 2024.

**Experimental Designs Or Analyses:**

The actual design and analyses seem fine but I find them insufficient, see "Methods And Evaluation Criteria".

**Methods And Evaluation Criteria:**

The MEC method makes a lot of sense as an alternative to OT for the problem of multimodal alignment, which solves the problem of the marginals being in different spaces trivially. I can't really comment on whether the diffusion approach makes sense as I am unfamiliar with MEC, but it seems like a reasonable approach.

The evaluation leaves much to be desired, unfortunately. The single-cell dataset from the SCOT paper appears to be a poor choice for demonstrating the strengths of the MEC method, as SCOT already performs so well in this case. Another single-cell dataset with ground-truth pairings available is the CITE-seq dataset from the NeurIPS benchmark from 2022 that many other works in this area also evaluate on.

The image translation dataset makes little sense to include as it is not really multimodal, and the authors do little to support why MEC would be preferable to existing methods such as conditional diffusion or CycleGAN. Note the authors write (l373)
> By design, our method does not require comparable domains and does not rely on a specific image similarity measure.
Which I agree is a strength but is not demonstrated through the experiment itself. As a suggestion, it would be much more interesting to look at image--caption pairs (which are abundant), artificially split them, and then evaluate on this instead.

**Other Comments Or Suggestions:**

### Typos/unclear points

- See also "Theoretical Claims".

- Eqn 3 has $E_{y \sim p_Y}[\mathbb{H}(p_{X\mid Y})]$. Is this a typo? I think it should probably be $\mathbb{H}(P_{x,y}) = E_{x,y}[-\log p_{X \mid Y}^\theta - \log p_y]$, the latter term is a constant for Eqn 4.

- I am trying to wrap my head around Eqn 6. How does this find a MEC? Doesn't $E_{x, y \sim P_{X,Y}^\theta}( -\log p_{Y\mid X}^\phi(y \mid x))$ control the __relative__ entropy (ie KL divergence) instead of the actual entropy? I can kind of understand how this enforces $P_{X,Y}^\theta = P_{X,Y}^\phi$, since
$$
\mathbb{KL}(P_{X,Y}^\theta \mid P_{X,Y}^\phi) = E_{x, y \sim P_{X,Y}^\theta}( -\log p_{Y\mid X}^\phi(y \mid x) - \log p_X^\phi(x)) = E_{x, y \sim P_{X,Y}^\theta}( -\log p_{Y\mid X}^\phi(y \mid x)) + \mathbb{KL}(P_{X}^\theta \mid P_{X}^\phi)
$$
but I don't understand how it gives a MEC.

- In Eqn. 11 you compute the marginal $p_X^\theta$ by $p_{X \mid Y}(x \mid y = \emptyset)$. However there is a true marginal corresponding to your model $p_X^\theta = \int_{y} P_{X \mid Y=y}(x \mid y) dy$. Is the approach in Eqn. 11 a reasonable approximation to the marginal?

**Other Strengths And Weaknesses:**

### Strengths

- The paper does great job reviewing the literature ranging from the problem, applications and solutions, as well as effectively introducing the MEC problem. The writing was clear and approachable.

### Weaknesses

- I actually think the paper undersells the relevance of MEC for this problem, since it completely avoids the problem of different modalities. In Section 4.1 the authors briefly mention

>All alternative methods we consider require geometric distances or similarity measures, which is a pain point that our method DDMEC lifts completely.

In my opinion this should absolutely be a central selling point of the method.

- The authors do not support the usage of MEC for alignment theoretically nor intuitively. Why should we expect minimum entropy to be a good objective that aligns unpaired data? What are your assumptions about how these data became unpaired in the first place?

- There is no theoretical support for the DDMEC itself: the maximum likelihood part seems fine to me but I am not sure about the alternating optimization.

**Questions For Authors:**

I really like the idea of using MEC to solve this problem (see weakness 1 above). However the paper falls short on two counts. I would be happy to argue for acceptance (at least, weak accept) if satisfactory improvements/answers to both of these were made.

### (1) Experiments

See "Methods And Evaluation Criteria"

### (2) Technical Details

See "Other Comments Or Suggestions".

**Relation To Broader Scientific Literature:**

This is an important problem that has been especially significant for single-cell biology, where measurement processes can usually only be taken once per cell, but different measurements can nonetheless capture additional information. The authors do a good job of reviewing the literature here.

**Theoretical Claims:**

The paper makes few theoretical claims. There is a claim on l190 under Equation 6 about the approximate equivalence of gradients that has no justification, and it is not clear to me that it would be true (these are conditional distributions over completely different spaces).

Also, is there a typo here? $-\log p_{Y \mid Y}$ should maybe be $-\log P_{Y \mid X}$, and is the gradient on the RHS supposed to be w.r.t. $\theta$ or $\phi$?)

---

> ### Author Rebuttal · Authors · 2025-03-31
>
> Thank you for recognizing the relevance of the MEC framework in coupling unpaired data and for the insightful feedback, which we address next.
>
> > Another single-cell dataset.
>
> We perform new experiments using the peripheral blood mononuclear cells (PBMC) dataset, as the CITE-seq dataset is used in a semi-supervised setting: while we can ignore label information with DDMEC, we lack the time to run competitors. The PBMC dataset is large and high-dimensional, consisting of simultaneous profiling of gene expression (RNA) and chromatin accessibility (ATAC) of healthy patients (refer to question 1 of Reviewer 2cNi). Our new results indicate that OT fails likely due to issues related to data dimensionality, while DDMEC achieves the best performance in all cases.
>
> > Image translation is not really multimodal. Use image caption pairs.
>
> Both modalities are images though the unpaired setting is a significant challenge. CycleGAN lags behind DMs. DDmec outperforms DM alternatives without the need to define a similarity function like in EGSDE and SDDM ( additional CelebA experiments in RV 2cNi question 1). While interesting, due to time limits, we differ the suggested txt/img case to a camera ready version.
>
> > The claim on l190 has no justification.
>
> We apologize, $- \log p_{Y|Y}$ is a typo and should be $- \log p_{Y|X}$.
>
> The l.h.s term of l190 reads $\nabla_\theta \int p_Y(y) p^\theta_{X|Y}(x|y)  \log p^\theta_{X|Y}(x|y)dx dy$. Moving the $\nabla_\theta$ inside we obtain
>
> $$ \int p_Y(y) \nabla_\theta\left(p^\theta_{X|Y}(x|y)\right)  \log p^\theta_{X|Y}(x|y)dx dy +  \int p_Y(y) p^\theta_{X|Y}(x|y)  \nabla_\theta\left(\log p^\theta_{X|Y}(x|y)\right) dx dy
> $$
>
> The second term simplifies to zero:
> $$ \int p_Y(y) p^\theta_{X|Y}(x|y)  \nabla_\theta\left(\log p^\theta_{X|Y}(x|y)\right) dx dy=
> \int p_Y(y) \nabla_\theta\left(p^\theta_{X|Y}(x|y)\right) dx dy=\nabla_\theta\int p_Y(y)\left(p^\theta_{X|Y}(x|y)\right) dx dy=\nabla_\theta1=0  $$
>
> Assuming $p^\theta_{X,Y}=p^\phi_{X,Y}$ and $p^\theta_X=p_X,p^\phi_Y=p_Y$, the first term rewrites as
> $$
> \int p_Y(y) \nabla_\theta\left(p^\theta_{X|Y}(x|y)\right)  \log \frac{p^\phi_{Y|X}(y|x) p_X(x)}{p_Y(y)} dx dy =  \int p_Y(y) \nabla_\theta\left(p^\theta_{X|Y}(x|y)\right)  (\log p^\phi_{Y|X}(y|x) +\log p_X(x)-\log p_Y(y)) dx dy=
> $$
> $$
> = \nabla_\theta\int p_Y(y)\left(p^\theta_{X|Y}(x|y)\right)  \log p^\phi_{Y|X}(y|x)
> $$
> which is the rhs of l190. Indeed $\int p_Y(y) \nabla_\theta\left(p^\theta_{X|Y}(x|y)\right) \log p_X(x) dx dy=\int \nabla_\theta\left(p^\theta_{X}(x)\right) \log p_X(x) dx=\int \nabla_\theta\left(p_{X}(x)\right) \log p_X(x) dx=0$, and similarly, $-\int p_Y(y) \nabla_\theta\left(p^\theta_{X|Y}(x|y)\right) \log p_Y(y) dx dy=0$.
>
> > Clarify Eqn 3 and Eqn 4.
>
> To clarify, we recognize that
> $$
> H(p^\theta_{X,Y})=-\int p^\theta_{X,Y}(x,y) \log p^\theta_{X,Y}(x,y) dx dy=-\int p_Y(y)p^\theta_{X|Y}(x|y)  \left( \log p^\theta_{X|Y}(x|y)+ \log p_Y(y) \right) dx dy=
> $$
> $$
> -\int p_Y(y)p^\theta_{X|Y}(x|y)  \left( \log p^\theta_{X|Y}(x|y) \right) dx dy+H(p_Y).
> $$
>
> The term $H(p_Y)$ does not influence optimization being independent of $\theta$, while the first term, can be either rewritten as $E_{y\sim p_Y}[H(p_{X| Y=y}) ]$ (as in eqn 3) or as $- E_{x,y\sim p_{X,Y}}[\log p_{X|Y}(x|y)]$, which is the expression in eqn 4.
>
> > How does Eqn 6 finds a MEC?
>
> To see that eqn 6 is a proxy for a MEC loss, we start from equations 4 and 5, which correspond exactly to two (uncoupled) MEC problems. We combine them in a system where we use the assumption from line 190, which allows swapping the loglikelihood terms: from the perspective of a gradient based optimizer, two losses which induce the same gradients are equivalent. Attention: we need to enforce the joint constraint throughout training, as discussed in L 241.
>
> > Is the approach in Eqn. 11 a reasonable approximation to the marginal?
>
> We apologize, the term $\epsilon^\theta(x_t, y = \emptyset, t))$ is a typo and should be $\epsilon^\theta(x_t, y, t)$. To enforce the marginal constraint, we keep the conditional model close to the frozen unconditional model by following Fan et al. (2023), which upper-bounds the KL divergence (see their Equation 6). In their work, fine-tuning preserves proximity to the original model, whereas we learn a conditional model while maintaining closeness to the frozen unconditional model. Our Equation 11 reformulates Equation 6 from (Fan et al,2023) in terms of denoisers.
>
> > Why MEC is a good objective that aligns unpaired data?
>
> The MEC problem is well-established and grounded in the domain of information theory. By minimizing entropy while satisfying marginal constraints, MEC maximizes mutual information, capturing shared structure between distributions. Unlike optimal transport, which relies on a predefined ground metric and may overlook structural relationships, MEC aligns unpaired data without requiring an explicit similarity measure.

---

> > ### Comment · Reviewer_Acbm · 2025-04-09
> >
> > I see, the derivation depends on the exact constraints being satisfied and you do disclose the approximate nature of the expression in the paper. Thank you. This derivation should be included in (an Appendix of) the paper.
> >
> > On the image translation experiments, I now realize the authors don't exclusively frame their method as only applicable to the multimodal setting. OK, this was probably my own bias as someone more interested in the multimodal problem. Image translation makes sense as an evaluation then. The PBMC experiment is nice, but again I think it fails to showcase the most exciting aspect of the method which is that it handles multimodality so long as separate generative models can be trained on each modality. Since the PBMC dataset is dimension reduced first it does not showcase this but I recognize the time constraints here.
> >
> > After reading the response and further reading of the paper, **I think this work makes a lot of interesting contributions, and so I will raise my score to 4**. However I feel the presentation has a high risk of being under-appreciated and I hope the authors can take some of my suggestions into account:
> >
> > - Currently, the paper is written heavily emphasizing multimodality, which leads to disappointment that the more impressive experiment is on a uni-modal image translation task. My suggestion is to frame MEC as a novel approach to learning unpaired **continuous** data that avoids the distance calculation in both high dimensional **and** multimodal settings, as long as good generative models can be learned.
> >
> > - The ability to do conditional sampling should be emphasized more. In multimodal settings, computationally feasible OT approaches can only sample over the empirical distribution of the data, but your method can sample genuinely novel instances (eg Figure 1). This seems commonplace in image translation but I think is potentially a substantial contribution in the biological settings. For example the fact that your method gets good FOS matching scores despite generating novel data and needing to select a nearest sample is impressive.

---

> > > ### Author Response · Authors · 2025-04-09
> > >
> > > Dear Reviewer Acbm,
> > > thank you very much for your additional feedback and advice, which is very useful and that we will follow in our revised paper.
> > >
> > > Thanks,
> > > the Authors

---

### Official Review · Reviewer_zrWS · 2025-03-14

**Overall Recommendation:** 3

**Summary:**

The manuscript presents a novel method for matching unpaired data through Minimum Entropy Coupling (MEC). By extending MEC to continuous distributions and leveraging denoising diffusion probabilistic models (DDPMs), the authors propose a cooperative framework that alternates between two conditional generative models. The method is evaluated on single-cell multi-omics alignment and unpaired image translation. Experimental results show competitive performance compared to existing state-of-the-art approaches.

**Claims And Evidence:**

The claim that a cooperative diffusion-based approach can be used to solve the continuous MEC problem is supported by both theoretical derivations and experiments. The author claim that DDMEC is applicable across diverse domains. Although the experiments on two tasks are promising, these evaluations may not be sufficient to claim full generality. More extensive testing on a broader range of datasets or modalities could better support this claim. Claims that the proposed method outperforms state-of-the-art methods are supported by quantitative results. However, the margins of improvement is relatively small, and no repeated experiments are conducted to measure the stability of the method.

**Essential References Not Discussed:**

No.

**Experimental Designs Or Analyses:**

The experimental designs and analyses are fundamentally sound. The authors did not provide sensitivity analyses, and computational cost discussion. No repeated experiments are conducted to measure the stability of the method.

**Methods And Evaluation Criteria:**

The methods and evaluation criteria make sense for the problem at hand. The authors propose method is well aligned with the challenge of matching unpaired data in multimodal settings.
The choice of benchmark datasets and metrics also appears appropriate.

**Other Comments Or Suggestions:**

No.

**Other Strengths And Weaknesses:**

The paper makes contributions by extending the MEC framework from discrete to continuous settings.
The authors provide derivation of the optimization objectives, including soft marginal constraints and the joint cooperative scheme.
Experimental results on two distinct tasks demonstrates the method’s performance

The manuscript would benefit from more detailed discussion and analysis on how sensitive the method is to these choices could strengthen the empirical section.
A discussion of computational cost and runtime compared to baseline methods would help assess the practicality of the approach.
Repeated experiments could be helpful in measuring the stability of the method.

**Questions For Authors:**

1. Could you provide a more detailed discussion and analysis on the sensitivity of your method to various hyperparameter choices?

2. Could you elaborate on the computational cost and runtime of your approach compared to baseline methods?

3. Have you conducted repeated experiments to measure the stability of your method?

**Relation To Broader Scientific Literature:**

This paper extends the Minimum Entropy Coupling (MEC) framework to continuous distributions. By reinterpreting the joint distribution as two conditional generative models, the work is related to generative diffusion models.

**Theoretical Claims:**

The proofs and the theoretical claims appear correct.

---

> ### Author Rebuttal · Authors · 2025-03-31
>
> We thank the reviewer for recognizing the contribution of our method for matching unpaired data and the insightful feedback which we address in the following. As suggested by the other reviewers as well, we performed additional experimental campaigns to challenge our DDMEC method.
>
> > 1- Could you provide a more detailed discussion and analysis on the sensitivity of your method to various hyperparameter choices?
> The hyperparameters on which our method depends are:
>
> * **Classifier Guidance Weight**: This parameter acts as a temperature scale that determines the influence of conditioning in the diffusion model. Since our reinforcement learning framework for diffusion models requires sampling from the model during training, we use classifier guidance and set it to cfg = 7, following (Fan,2023). In the plot available here [ https://anonymous.4open.science/r/icml2025_ddmec-7798/celeba/ablations.png ], we show the results of a new ablation study on the guidance scale in the unpaired image translation case, which is performed at test time. The plot displays the trade-off between FID and SSIM scores, as a function of the guidance scale for two different number of sampling steps (50 and 100).
> Intuitively, we observe that the guidance scale controls the tradeoff between information transfer between modalities and fidelity to the marginal constraint. A higher guidance scale improves SSIM but worsens FID, while a lower guidance scale favors FID but negatively impacts SSIM. Under the MEC framework, guidance allows us to balance the maximization of mutual information (by lowering the joint entropy) and maintaining fidelity to the marginals.
>
> - **Lambda Weight**: We find that a low KL divergence weight is essential for ensuring reward convergence, thus set
>  $\hat{\lambda_X} = \hat{\lambda_Y} = 0.001$, in equation (9) for all experiments. Indeed, as noted in (Fan, 2023), the  $\hat{\lambda_{(\cdot)}}$ parameter helps the diffusion model to remain close to the unconditional model and prevents reward overfitting, thus avoiding the degenerate solution where the marginal constraints are ignored. Recall that our method initializes the model with the unconditional model, so we start from a state where the marginal constraints are fully respected and the KL divergence is near zero. Thus, the role of lambda is weight to the KL to keep the conditional model close to the unconditional one while optimizing the joint entropy coupling.
>
> Fan, Y., Watkins, O., Du, Y., Liu, H., Ryu, M., Boutilier, C., ... & Lee, K. Dpok: Reinforcement learning for fine-tuning text-to-image diffusion models. Neurips2023.
>
> > 2 - Could you elaborate on the computational cost and runtime of your approach compared to baseline methods?
>
> We discuss computational costs of our and baseline methods according to an application-specific division.
>
> - **Image Translation**: Given that diffusion models (DMs) outperform GANs for this task, we compare DDMEC against other diffusion-based methods. The inference process of our method behaves similarly to an unconditional model, with additional conditional layers to incorporate guidance. The generation requires an iterative process with a guidance mechanism. We find that 50 steps are sufficient to achieve superior performance. This is significantly more efficient than SDEdit and EGSDE, which require 1000 steps, and SDDM and ILVR, which require 100 steps.
>
>
> -  **Single-Cell RNA (ScRNA) Analysis**: For the SNARE dataset, optimal transport (OT) methods are computationally lightweight and achieve results comparable to DDMEC. However, this dataset is limited, with dimensionalities of 19 and 10 for the two modalities and only 1,047 samples. To further assess scalability, we conducted additional experiments on larger and high-dimensional datasets such as PBMC (see Question 1 of Reviewer 2cNi). The results indicate that OT methods fail entirely on this dataset, whereas alternative methods like DDMEC and ScTopoGAN perform significantly better. In general, approaches to solve the OT problem in high dimensions face challenges related to the cost matrix computation and fundamental difficulties of distance estimation in high dimensions.
>
> Comparison with ScTopoGAN: In terms of computational time, DDMEC is comparable to ScTopoGAN. However, ScTopoGAN involves a first topological autoenconding and a GAN model trained via multiple distinct phases and multiple runs to select the best model over multiple seeds, making a direct computational comparison challenging.
>
> >3 - Have you conducted repeated experiments to measure the stability of your method?
>
> Inference-time results are run over several seeds to improve statistical significance. Our new results show confidence intervals in terms of standard deviation.

---

### Decision · Program_Chairs · 2025-05-01

**Decision:**

Accept (poster)

**Comment:**

This paper proposes  method for alignment of continuous  multimodal data using Minimum Entropy Coupling. Reviewers have appreciated the contribution and its usefulness in real-world scenario. I recommend to accept the paper while requesting the authors to incorporate the reviews comments and experiments that came up during the rebuttal.